# One-step fermentative production of aromatic polyesters from glucose by metabolically engineered *Escherichia coli* strains

Jung Eun Yang[1], Si Jae Park[2], Won Jun Kim[1], Hyeong Jun Kim[3], Bumjoon J. Kim[3], Hyuk Lee[4], Jihoon Shin[5] & Sang Yup Lee [ORCID][1,6,7]

Aromatic polyesters are widely used plastics currently produced from petroleum. Here we engineer *Escherichia coli* strains for the production of aromatic polyesters from glucose by one-step fermentation. When the *Clostridium difficile* isocaprenoyl-CoA:2-hydroxyisocaproate CoA-transferase (HadA) and evolved polyhydroxyalkanoate (PHA) synthase genes are overexpressed in a D-phenyllactate-producing strain, poly(52.3 mol% 3-hydroxybutyrate (3HB)-*co*-47.7 mol% D-phenyllactate) can be produced from glucose and sodium 3HB. Also, various poly(3HB-*co*-D-phenyllactate) polymers having 11.0, 15.8, 20.0, 70.8, and 84.5 mol% of D-phenyllactate are produced from glucose as a sole carbon source by additional expression of *Ralstonia eutropha* β-ketothiolase (*phaA*) and reductase (*phaB*) genes. Fed-batch culture of this engineered strain produces 13.9 g l$^{-1}$ of poly(61.9 mol% 3HB-*co*-38.1 mol% D-phenyllactate). Furthermore, different aromatic polyesters containing D-mandelate and D-3-hydroxy-3-phenylpropionate are produced from glucose when feeding the corresponding monomers. The engineered bacterial system will be useful for one-step fermentative production of aromatic polyesters from renewable resources.

[1] Metabolic and Biomolecular Engineering National Research Laboratory, Department of Chemical and Biomolecular Engineering (BK21 Plus Program), Center for Systems and Synthetic Biotechnology, Institute for the BioCentury, Korea Advanced Institute of Science and Technology (KAIST), Daejeon 34141, Republic of Korea. [2] Division of Chemical Engineering and Materials Science, Ewha Womans University, Seoul 03760, Republic of Korea. [3] Polymer and Nano Electronics Laboratory, Department of Chemical and Biomolecular Engineering (BK21 Plus Program), Institute for the BioCentury, KAIST, Daejeon 34141, Republic of Korea. [4] Division of Drug Discovery Research, Korea Research Institute of Chemical Technology, Daejeon 34114, Republic of Korea. [5] Center for Bio-based Chemistry, Green Chemistry & Engineering Division, Korea Research Institute of Chemical Technology, Daejeon 34114, Republic of Korea. [6] BioProcess Engineering Research Center, KAIST, Daejeon 34141, Republic of Korea. [7] BioInformatics Research Center, KAIST, Daejeon 34141, Republic of Korea. Jung Eun Yang and Si Jae Park contributed equally to this work. Correspondence and requests for materials should be addressed to S.Y.L. (email: leesy@kaist.ac.kr)

Aromatic polyesters are widely used indispensable plastics currently produced from petroleum[1]. Microbial fermentative production of polymers from renewable resources has received much attention to substitute petroleum-based polymers and help solving environmental problems. Thus, there has been much interest in fermentative production of aromatic polyesters from renewable non-food biomass, but without any success. Polyhydroxyalkanoates (PHAs) comprising various hydroxycarboxylic acids are natural biodegradable microbial polyesters, which are accumulated by numerous microorganisms under nutrient limited conditions in the presence of excess carbon sources[2,3]. The material properties of PHAs can be modulated by changing the monomer types and compositions. Over the last three decades, numerous metabolic engineering studies have been performed to produce PHAs containing specific monomers having different monomer compositions. Various kinds of monomers such as 3-hydroxypropionate, 3-hydroxybutyrate (3HB), 3-hydroxyvalerate, 4-hydroxybutyrate, 5-hydroxyvalerate, and 6-hydroxyhexanoate, and medium-chain length 3-hydroxyalkanoates have been shown to be incorporated into PHAs, either as homo- or co-polymers, giving diverse material properties[2,4–7].

More recently, one-step fermentative production of non-natural polyesters, poly(D-lactate), poly(lactate-co-glycolate), and other D-lactate-containing PHAs by metabolically engineered bacteria have been reported[7–13]. In these studies, an evolved propionate CoA-transferase (Pct) and an evolved PHA synthase (PhaC) were expressed in *Escherichia coli*; engineered *Clostridium propionicum* $Pct_{cp}$ converts 2-hydroxyacids to corresponding 2-hydroxyacyl-CoAs using acetyl-CoA as a CoA-donor and then engineered *Pseudomonas* sp. MBEL 6–19 PHA synthase ($PhaC_{Ps6–19}$) polymerizes these 2-hydroxyacyl-CoAs into polyesters containing corresponding monomers[9–11]. It has previously been reported that some bacteria such as *Pseudomonas oleovorans* and *Pseudomonas putida* strains can synthesize aromatic polyesters when grown in the culture medium containing *n*-phenylalkanoic acid as a direct precursor of aromatic monomer[14–16]. These aromatic polyesters have only been produced by feeding the cells with corresponding aromatic monomers as substrates and have not been produced by direct fermentation from renewable feedstock carbohydrates such as glucose. Furthermore, those mentioned above contain aromatic groups far away from the main polymer carbon chain[14–16] and thus polymer properties were found to be much different from petroleum-derived aromatic polymers, which often contain aromatic rings close to the main polymer chain, such as poly(ethylene terephthalate) (PET) and polystyrene. Such observations led us to develop metabolically engineered *E. coli* strains for one-step fermentative production of aromatic polyesters from glucose.

The following strategies were employed in this study. First, CoA-transferases that can efficiently activate phenylalkanoates into their corresponding CoA derivatives were discovered. Second, cells were metabolically engineered to produce phenylalkanoates from glucose. Third, the engineered phenylalkanoates overproducing *E. coli* strains were employed for in vivo production of aromatic polyesters by expressing engineered PHA synthase and CoA-transferases. Fourth, strains were further engineered to produce aromatic polyesters directly from glucose. Fifth, the enzyme expression levels were modulated to produce various aromatic polyesters having different monomer fractions. Finally, as proof-of-concept examples of expanding the range of aromatic polyesters that can be produced by fermentation, aromatic polyesters containing D-mandelate and D-3-hydroxy-3-phenylpropionate were produced by feeding the corresponding precursors.

## Results

**Validation of the CoA-transferase activity**. First, we examined whether Pct can activate phenyllactate and mandelate into phenyllactyl-CoA and mandelyl-CoA, respectively. As the mutant of $Pct_{cp}$ (Pct540) has successfully been employed for in vivo production of polyesters containing 2-hydroxyacids such as glycolate, lactate, 2-hydroxybutyrate, and 2-hydroxyisovalerate, and various hydroxyacids[7], the substrate spectrum of Pct540 seems to be broad enough with respect to the number of carbon atoms and position of hydroxyl group. Unfortunately, however, Pct540 was found to have no catalytic activities toward phenyllactate and mandelate (Supplementary Fig. 1). Thus, a CoA-transferase capable of activating aromatic compounds into corresponding CoA derivatives had to be identified for aromatic polymer production.

There have been reports describing that *Clostridium sporogenes* cinnamoyl-CoA:phenyllactate CoA-transferase (FldA) can convert phenyllactate into phenyllactyl-CoA using cinnamoyl-CoA as a CoA-donor[17]. As cinnamoyl-CoA is a non-natural metabolite in *E. coli*, the possibility of using acetyl-CoA, which is abundant metabolite in the cell, as a CoA-donor was examined for the synthesis of phenyllactyl-CoA by *Clostridium botulinum* A str. ATCC 3502 FldA (using the synthesized gene), which shared 99.0% of amino acid sequence identity with that of *C. sporogenes*. However, FldA was found to have no catalytic activity on generating phenyllactyl-CoA using acetyl-CoA as a CoA-donor. A previous study reported that *Streptomyces coelicolor* 4-coumarate:CoA ligase (4CL) has an important role in phenylpropanoid metabolism *via* generating cinnamoyl-CoA from cinnamate[18]. Thus, a biosynthetic route was designed and tested in vitro for the synthesis of cinnamoyl-CoA from cinnamate by 4CL (Supplementary Fig. 2 and Supplementary Note 1). A modified 4CL[18] was applied to make cinnamoyl-CoA, which can be used as CoA-donor for FldA mediated phenyllactyl-CoA formation. As expected, phenyllactyl-CoA was successfully synthesized by in vitro sequential reaction of 4CL and FldA; 4CL converts cinnamate to cinnamoyl-CoA and then FldA converts phenyllactate into phenyllactyl-CoA (Supplementary Fig. 3). These results suggest that 4CL and FldA can potentially be used for phenyllactyl-CoA generation and consequently aromatic polyester production. Similarly, 4-hydroxyphenyllactate, another promising aromatic monomer, was also converted to 4-hydroxyphenyllactyl-CoA by in vitro sequential reaction of mutant 4CL and FldA (Supplementary Fig. 3).

**In vivo polymerization of aromatic PHA using FldA**. In the biosynthesis of non-natural polyesters, selection of a monomer-specific PhaC variant is crucial for the production of desired polyesters. To examine the performance of various PhaCs on aromatic PHAs production, engineered *Pseudomonas* sp. MBEL 6–19 PHA synthase ($PhaC_{Ps6–19}$) variants (PhaC1202, PhaC1301, PhaC1310, PhaC1437, and PhaC1439)[10] were expressed in *E. coli* XL1-Blue strain together with 3-deoxy-D-arabino-heptulosonate-7-phosphate (DAHP) synthase (AroG$^{fbr}$), phenylalanine ammonia-lyase (PAL), 4CL, FldA, and Pct540 in MR medium (see Methods) containing 20 g l$^{-1}$ (111 mM) of glucose, 1 g l$^{-1}$ of D-phenyllactate, and 1 g l$^{-1}$ of sodium 3HB. Here, 3HB was supplied to enhance the production of polymers, because it is converted by Pct540 into 3HB-CoA, a more favorable substrate of PhaC, thus allowing production of PHAs to sufficient amounts for further analysis. The *E. coli* XL1-Blue strains expressing different PHA synthase variants together with AroG$^{fbr}$, PAL, 4CL, FldA, and Pct540 were able to produce varying amounts of random copolymers, poly(D-lactate-co-3HB-co-D-phenyllactate), having different monomer compositions (Supplementary Fig. 4a).

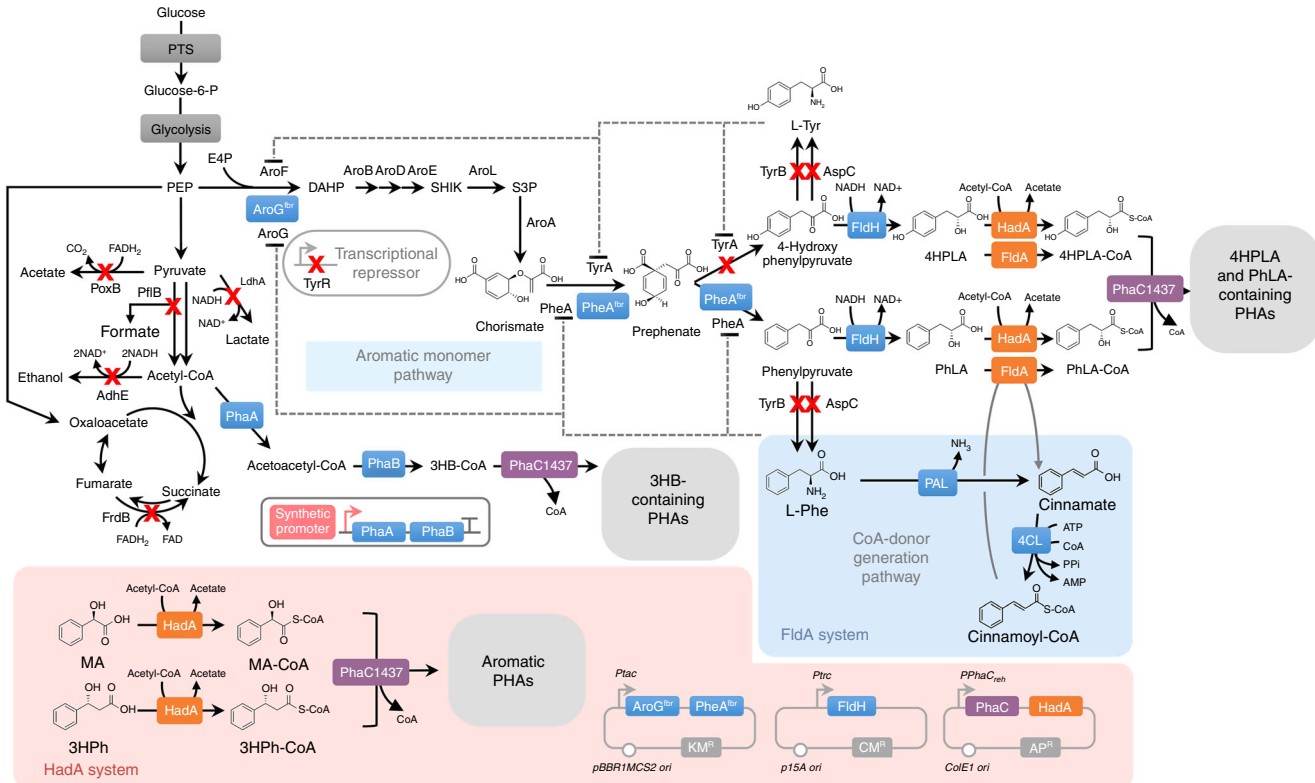

**Fig. 1** Metabolic engineering of *E. coli* for the production of aromatic PHAs. The overall strategies for the production of aromatic PHAs are shown. Blue boxes represent heterologous and endogenous enzymes introduced for the construction of metabolic pathways for aromatic PHAs. The orange and purple boxes indicate CoA-transferase and PHA synthase, respectively. The inactivated metabolic pathways are indicated by "X". Dotted lines indicate feedback inhibition. DAHP, 3-deoxy-D-arabino-heptulosonate-7-phosphate; E4P, erythrose-4-phosphate; 3HB-CoA, 3-hydroxybutyryl-CoA; 3HPh, D-3-hydroxy-3-phenylpropionate; 3HPh-CoA, D-3-hydroxy-3-phenylpropionyl-CoA; 4HPLA, D-4-hydroxyphenyllactate; 4HPLA-CoA, D-4-hydroxyphenyllactyl-CoA; L-Phe, L-phenylalanine; L-Tyr, L-tyrosine; PhLA, D-phenyllactate; MA, D-mandelate; MA-CoA, D-mandelyl-CoA; PhLA-CoA, D-phenyllactatyl-CoA; PEP, phosphoenolpyruvate; SHIK, shikimate; S3P, shikimate-3-phosphate

Among the PhaC variants, PhaC1437 containing four amino acids substitutions (E130D, S325T, S477G, and Q481K) was found to be the best, which resulted in the production of poly(18.3 mol% D-lactate-*co*-76.9 mol% 3HB-*co*-4.8 mol% D-phenyllactate) to the polymer content of 7.8 wt% of dry cell weight (Supplementary Fig. 4a, b).

As externally supplemented D-phenyllactate was successfully incorporated into PHAs, it was aimed to engineer *E. coli* to generate D-phenyllactate in vivo from glucose directly. Biosynthesis of aromatic compounds starts from DAHP, which is derived from the condensation of phosphoenolpyruvate and erythrose-4-phosphate by DAHP synthase. The DAHP is converted into phenylpyruvate and then further converted to D-phenyllactate by D-phenyllactate dehydrogenase (FldH) (Fig. 1). It is well known that the metabolic pathways for the biosynthesis of aromatic compounds are regulated by various inhibitory mechanisms[19]; expression of DAHP synthase encoded by *aroG* and chorismate mutase/prephenate dehydratase encoded by *pheA* are inhibited by L-phenylalanine[20]. To produce D-phenyllactic acid in vivo, negative feedback inhibition resistant mutants AroG^fbr [AroG (D146N)] and PheA^fbr [PheA (T326P)] were constructed based on the previous reports[21,22]. The engineered *E. coli* XL1-Blue strain expressing AroG^fbr, PheA^fbr, and the *C. botulinum* A str. ATCC 3502 FldH was able to produce 0.372 g l⁻¹ of D-phenyllactic acid from 15.2 g l⁻¹ (84.4 mM) of glucose. Moreover, additional overexpression of PAL, 4CL, FldA, Pct540, and PhaC1437 in this strain resulted in production of poly(16.8 mol% D-lactate-*co*-80.8 mol% 3HB-*co*-1.6 mol% D-phenyllactate-*co*-0.8 mol% D-4-hydroxyphenyllactate) to the polymer content of 12.8

wt% of dry cell weight in MR medium containing 20 g l⁻¹ of glucose and 1 g l⁻¹ of sodium 3HB (Supplementary Fig. 5 and Supplementary Table 1). It should be noted that a small amount of D-4-hydroxyphenyllactate was also incorporated into the polymer, which is predictable from in vitro FldA assay (Supplementary Fig. 3); D-4-hydroxyphenyllactyl-CoA was also generated by FldA and polymerized with D-phenyllactyl-CoA.

**Identification of 2-hydroxyisocaproate CoA-transferase.** Although aromatic PHAs containing D-phenyllactate could be successfully produced, critical problems still exist in using the FldA system, such as rather low aromatic monomer content and narrow monomer spectrum. In vitro enzyme assay results suggested that FldA was able to transfer CoA to phenyllactate and 4-hydroxyphenyllactate, but not to others of our interest, such as mandelate, 2-hydroxy-4-phenylbutyrate, 3-hydroxy-3-phenyl-propionate, and 4-hydroxybenzoate (Supplementary Fig. 3). Thus, the *Clostridium difficile* isocaprenoyl-CoA:2-hydro-xyisocaproate CoA-transferase (HadA), which has more than 48% of amino acid sequence identity to FldA was newly identified based on amino acid sequence similarity analysis (Supplementary Fig. 6).

Interestingly, despite isocaprenoyl-CoA-based 2-hydroxyiso-caproate-specific CoA transferring activity of HadA[23], we newly discovered that phenyllactate could be converted to phenyllactyl-CoA using acetyl-CoA derived CoA moiety (Fig. 2 and Supplementary Fig. 7). Moreover, HadA showed catalytic activity on converting mandelate, 4-hydroxymandelate, 4-

hydroxyphenyllactate, 2-hydroxy-4-phenylbutyrate, 3-hydroxy-3-phenylpropionate, and 4-hydroxybenzoate to their corresponding CoA derivatives (Fig. 2 and Supplementary Fig. 7c–k). Thus, HadA has a potential to more efficiently produce diverse aromatic polyesters using acetyl-CoA as a CoA donor.

**Enhanced carbon flux for D-phenyllactic acid production.** For the overproduction of D-phenyllactate, the *tyrR* gene was deleted to make *E. coli* XBT strain due to the negative regulation of TyrR on aromatic amino acid biosynthesis. *E. coli* XBT strain expressing AroG^fbr, PheA^fbr, and FldH could produce 0.5 g l⁻¹ of D-

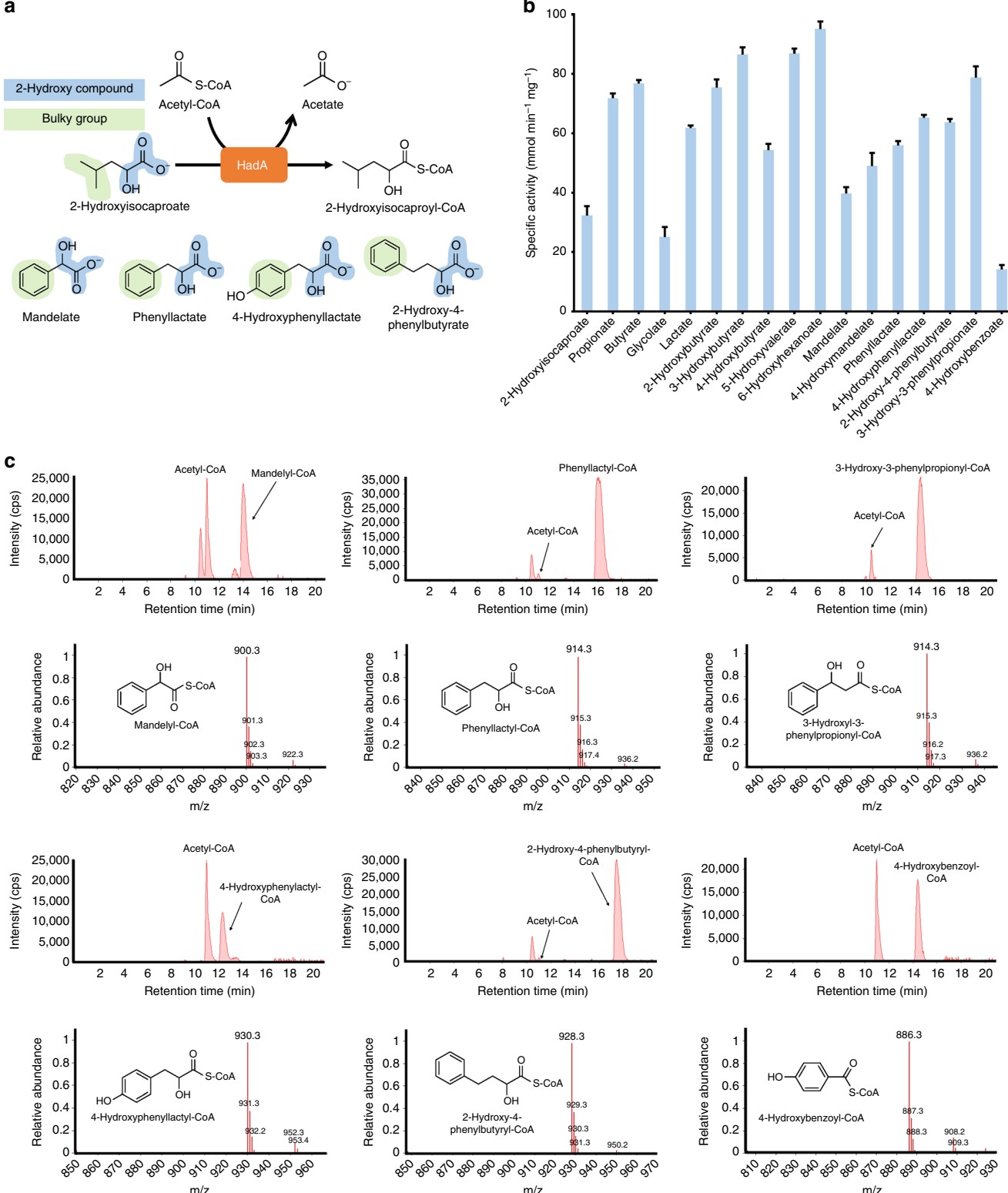

**Fig. 2** Activity of HadA on various substrates in vitro. **a** Schematic representation of reaction used in the assay. **b** The specific activities of HadA on various substrates. Data represent mean ± SD (*n* = 3 technical replicates). **c** LC-MS analysis of various aromatic acyl-CoAs generated by HadA during the in vitro assay

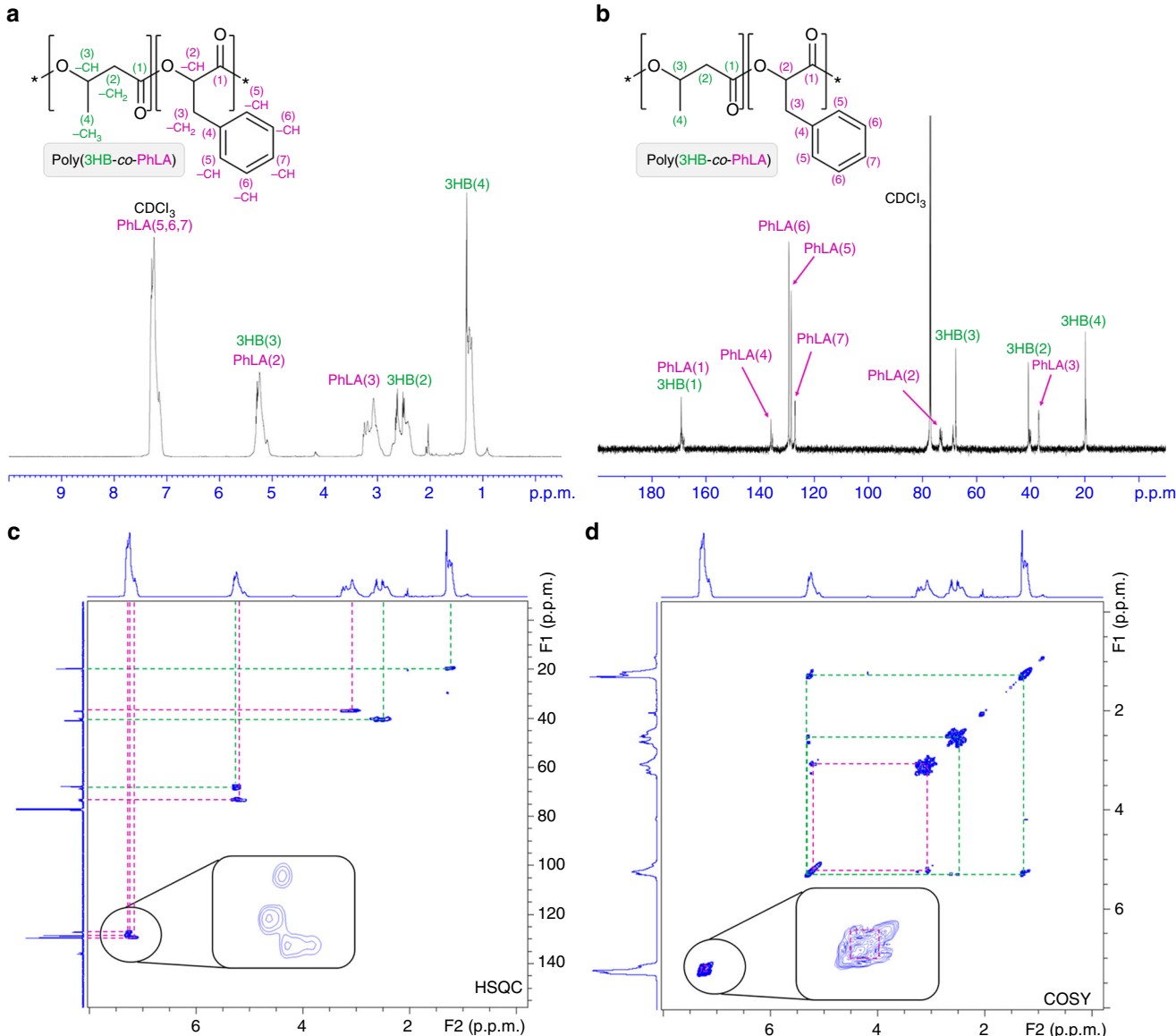

**Fig. 3** Characterization of aromatic PHAs produced by metabolically engineered *E. coli*. **a–d** NMR spectra of poly(3HB-*co*-D-phenyllactate) produced in *E. coli* XB201TBAL expressing AroG^fbr, PheA^fbr, FldH, HadA, and PhaC1437. **a** ¹H NMR. **b** ¹³C NMR. **c** 2D ¹H-¹³C HSQC NMR. **d** 2D ¹H-¹H COSY NMR

phenyllactic acid from 16.4 g l⁻¹ (91 mM) of glucose, which is 34.4% higher than that obtained with *E. coli* XL1-Blue strain expressing the same genes. In order to remove competing pathways of D-phenyllactate biosynthesis, the *poxB* (encoding pyruvate oxidase), *pflB* (encoding pyruvate formate lyase), *adhE* (encoding acetaldehyde dehydrogenase/alcohol dehydrogenase), and *frdB* (encoding fumarate reductase) genes were deleted in *E. coli* XBT, to make *E. coli* XB201T. The *E. coli* XB201T strain expressing AroG^fbr, PheA^fbr, and FldH produced 0.55 g l⁻¹ of D-phenyllactic acid from 15.7 g l⁻¹ (87 mM) of glucose, which is 10% higher than that obtained with the XBT strain (Supplementary Fig. 8). The *tyrB* gene encoding tyrosine aminotransferase and *aspC* gene encoding aspartate aminotransferase were also deleted, to increase carbon flux toward D-phenyllactic acid based on in silico genome-scale metabolic flux analyses (Supplementary Note 2). The resulting *E. coli* XB201TBA strain expressing AroG^fbr, PheA^fbr, and FldH allowed significantly enhanced production of D-phenyllactic acid up to 1.62 g l⁻¹ from 18.5 g l⁻¹ (102 mM) of glucose, which is 4.35-fold higher than that obtained with the original XL1-Blue strain expressing the same

genes. The *ldhA* gene was further deleted in XB201TBA to prevent D-lactate formation, to make XB201TBAL strain. The engineered *E. coli* XB201TBAL strain expressing AroG^fbr, PheA^fbr, FldH, HadA, and PhaC1437 was able to produce poly (52.1 mol% 3HB-*co*-47.9 mol% D-phenyllactate) to the polymer content of 15.8 wt% of dry cell weight in MR medium containing 20 g l⁻¹ of glucose and 1 g l⁻¹ of sodium 3HB (Fig. 3). It should be noted that D-4-hydroxyphenyllactate was not incorporated into the polymer anymore through reinforcing the flux toward D-phenyllactate. Moreover, poly(52.3 mol% 3HB-*co*-47.7 mol% D-phenyllactate) could be produced to a higher polymer content of 24.3 wt% by fed-batch fermentation (Supplementary Fig. 9).

The importance of HadA in the production of PHAs containing diverse aromatic monomers was also evaluated by supplementation of monomers of interest such as D-mandelate and D-3-hydroxy-3-phenylpropionate. The final engineered *E. coli* XB201TBAL expressing AroG^fbr, PheA^fbr, FldH, HadA, and PhaC1437 was able to produce poly(55.2 mol% 3HB-*co*-43.0 mol% D-phenyllactate-*co*-1.8 mol% D-mandelate) to the polymer content of 11.6 wt% of dry cell weight and poly(33.3 mol%

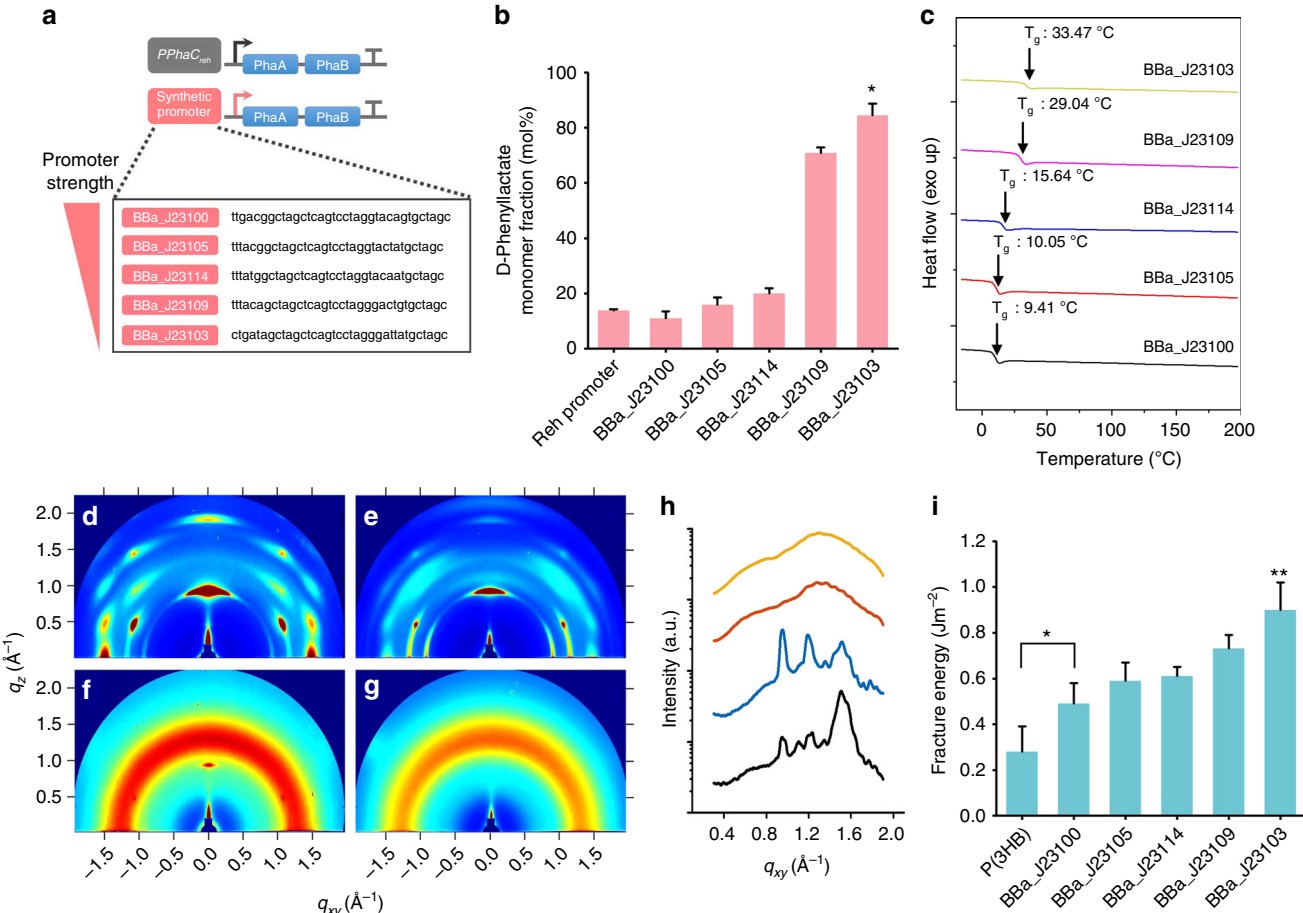

**Fig. 4** Characterization of synthetic promoter driven aromatic polyesters. **a** Scheme of the PhaAB expression cassette. Analysis of D-phenyllactate monomer fraction **b** and differential scanning calorimetry (DSC) analysis **c** of copolymers produced by XB201TBAL strain expressing PhaAB under the synthetic Anderson promoters (BBa_J23100, BBa_J23105, BBa_J23114, BBa_J23109, and BBa_J23103) followed by overexpression of AroG^fbr, PheA^fbr, FldH, HadA, and PhaC1437. **d–g** Grazing-incidence X-ray scattering (GIXS) patterns of poly(3HB) **d**, poly(16.8 mol% D-lactate-co-80.8 mol% 3HB-co-1.6 mol% D-phenyllactate-co-0.8 mol% D-4-hydroxyphenyllactate) **e**, poly(80.8 mol% 3HB-co-35.5 mol% D-phenyllactate) **f**, and poly(52.1 mol% 3HB-co-47.9 mol% D-phenyllactate) **g**. Colors indicate the intensity counts ranging from 0 (blue) to $10^5$ (red). **h** In-plane line cut of GIXS patterns of each polymer (**d** black; **e** blue; **f** orange; **g** yellow). **i** Dual cantilever beam (DCB) test of polymers. *$P < 0.05$ and **$P < 0.01$, by two-tailed $t$-test. Error bars represent s.d. of $n = 3$ technical replicates

3HB-co-18.0 mol% D-phenyllactate-co-48.7 mol% D-3-hydroxy-3-phenylpropionate) to the polymer content of 14.7 wt% of dry cell weight in MR medium containing 20 g l$^{-1}$ of glucose and 1 g l$^{-1}$ of sodium 3HB together with 0.5 g l$^{-1}$ of their corresponding monomers, respectively (Supplementary Fig. 10). These results strongly suggest that the engineered *E. coli* system expressing HadA and evolved PHA synthase can be broadly employed for the production of diverse aromatic polyesters.

**Synthetic promoter based flux modulation**. In the above studies, 3HB was supplemented to boost polymer accumulation as described earlier. Next, production of aromatic PHAs from glucose without 3HB supplementation was pursued in XB201TBAL strain by additionally expressing *Ralstonia eutropha* β-ketothiolase (PhaA) and acetoacetyl-CoA reductase (PhaB). As expected, XB201TBAL strain expressing AroG^fbr, PheA^fbr, FldH, HadA, and PhaC1437 was able to produce poly(86.2 mol% 3HB-co-13.8 mol% D-phenyllactate) to the polymer content of 18.0 wt% of dry cell weight in MR medium containing 20 g l$^{-1}$ of glucose as a sole carbon source. In addition, production of aromatic PHAs having different monomer fractions, which is important for industrial applications, was also attempted by modulating

metabolic fluxes of PhaAB using synthetic Anderson promoters (http://parts.igem.org/). Five different plasmids expressing PhaAB under five promoters of different strength were constructed and introduced into the XB201TBAL strain expressing AroG^fbr, PheA^fbr, FldH, HadA, and PhaC1437. The D-phenyllactate monomer fraction could be modulated, showing increased fraction with decreased PhaAB expression; polymers having 11.0, 15.8, 20.0, 70.8, and 84.5 mol% of D-phenyllactate could be produced (Fig. 4a, b and Supplementary Table 2); it should be noted that by expressing PhaAB under BBa_J23103 promoter, a polymer having a very high D-phenyllactate fraction (poly(15.5 mol% 3HB-co-84.5 mol% D-phenyllactate)) could be produced to the polymer content of 4.3 wt% of dry cell weight (Fig. 4b). These results suggest that aromatic polyesters having different aromatic monomer fractions can be produced by modulating metabolic fluxes.

Next, the pH-stat fed-batch culture of *E. coli* XB201TBAL strain expressing AroG^fbr, PheA^fbr, FldH, HadA, PhaC1437, and PhaAB under BBa_J23114 promoter was performed in a medium containing glucose without 3HB feeding. In 96 h, 2.5 g l$^{-1}$ of poly(67.6 mol% 3HB-co-32.4 mol% D-phenyllactate) was produced with a polymer content of 43.8 wt%, which demonstrated that aromatic polyester can be produced by one-step fermentation of

engineered *E. coli* from glucose (Supplementary Fig. 9). To further enhance production of aromatic PHAs, gene expression system was optimized by replacing the *ldhA* gene with the *fldH* gene in the chromosome of *E. coli* XB201TBA. In addition, the native promoter of the *ldhA* gene was replaced with the strong *trc* promoter to increase *fldH* expression. Furthermore, we employed pulsed-feeding method (see Methods). The resulting *E. coli* XB201TBAF strain expressing AroG$^{fbr}$, PheA$^{fbr}$, HadA, PhaC1437, and PhaAB under BBa_J23114 promoter allowed production of 13.9 g l$^{-1}$ of poly(61.9 mol% 3HB-*co*-38.1 mol% D-phenyllactate) with a polymer content of 55.0 wt% from glucose by fed-batch fermentation (Supplementary Fig. 9). This titer (13.9 g l$^{-1}$) obtained is 5.56-fold higher than that (2.5 g l$^{-1}$) obtained with the *E. coli* XB201TBAL strain expressing AroG$^{fbr}$, PheA$^{fbr}$, FldH, HadA, PhaC1437, and PhaAB under BBa_J23114 promoter and is also much higher than that (< 1 g l$^{-1}$) obtained by fed-batch culture of the *E. coli* XB201TBAL strain expressing AroG$^{fbr}$, PheA$^{fbr}$, FldH, HadA, and PhaC1437 in a medium supplemented with glucose and sodium 3HB. These results demonstrate that aromatic PHAs could be successfully produced to a reasonably high concentration by fed-batch culture of the engineered strain (XB201TBAF strain expressing AroG$^{fbr}$, PheA$^{fbr}$, HadA, PhaC1437, and PhaAB under BBa_J23114 promoter). Although we provided proof-of-concept fermentation results here, it is expected that further optimization of cultivation condition will allow more efficient production of aromatic polyesters by one-step fermentation from glucose.

Next, material properties of aromatic polyesters produced in this study were examined to see whether they are suitable for industrial applications. In the case of PET, the most widely used petroleum-derived aromatic polyester, fiber-grade, and bottle-grade PET have a number-average molecular weight ($M_n$; molecular mass) of 15–20 kDa and 24–36 kDa, respectively. The $M_n$'s of aromatic polyesters produced in this study are also in the range of 3.6–24.9 kDa (Supplementary Table 2), showing a good potential to replace petroleum-based aromatic polymers. To examine the material properties of the aromatic polyesters produced by metabolically engineered *E. coli* strains (Fig. 4c–i and Supplementary Note 3), films of five different aromatic polyesters produced above and poly(3HB) as a control were manufactured. The critical cohesive fracture energy ($G_c$) of poly(3HB) film was measured to be $0.28 \pm 0.11$ (SD) J m$^{-2}$, whereas the $G_c$ increased as the aromatic monomer fraction increased from 11.0 to 84.5 mol%. The polymer film made of poly(15.5 mol% 3HB-*co*-84.5 mol% D-phenyllactate), the polymer having the highest D-phenyllactate fraction, showed increased $G_c$ of $0.90 \pm 0.12$ (SD) J m$^{-2}$, which is mainly attributed to the reduced crystallinity and high glass transition temperature ($T_g$) of the polymer. These results suggest that aromatic polyesters produced by one-step direct fermentation of engineered *E. coli* strains can be used to replace those aromatic polyesters currently produced from petroleum.

## Discussion

In this study, we report the development of a bacterial platform system that allows production of various aromatic polyesters by one-step fermentation. The key strategies for the successful fermentative production of aromatic polyesters from glucose include identification and use of a novel broad substrate range CoA-transferase for activating aromatic compounds to their CoA derivatives and a mutant PHA synthase capable of polymerizing these aromatic CoA derivatives, and design and optimization of metabolic pathways to overproduce corresponding aromatic monomers in vivo. As demonstrated for several aromatic monomers, this system has a potential to be used for the

production of even more diverse aromatic polymers. For example, HadA (or related enzymes) and PHA synthase can be further engineered to accept desired aromatic monomers of interest. Recently, the crystal structure of *R. eutropha* PHA synthase has been determined[24,25]. This structure can serve as a model for rational protein engineering of various PHA synthases to broaden the substrate utilization range. It is expected that the bacterial platform system developed here will help establish sustainable bioprocesses for the production of aromatic polyesters from renewable non-food biomass to substitute petroleum-based counterparts.

## Methods

**Plasmids and construction of bacterial strains.** All chemicals used in this study were purchased from Sigma-Aldrich unless noted otherwise. All bacterial strains and plasmids used in this study were listed in Supplementary Table 3. *E. coli* XL1-Blue (Stratagene Cloning Systems, La Jolla, CA) was used for general gene cloning and all DNA manipulations were performed following the standard procedures[26]. PCR was performed with the C1000 Thermal Cycler (Bio-Rad, Hercules, CA). Primers used in this study (Supplementary Table 4) were synthesized by Genotech (Daejeon, Korea).

The pPs619C1437Pct540 was used for the expression of the *Pseudomonas* sp. 6–19 PHA synthase gene (*phaC*; ACM68707.1) containing quadruple mutations of E130D, S325T, S477G, and Q481K (PhaC1437), and the *C. propionicum* Pct gene containing mutation of V193A and four silent nucleotide mutations of T78C, T669C, A1125G, and T1158C (Pct540)[10].

To construct the pET22b-hisPCT540, the *pct540* gene was amplified by PCR using the primers Pcthis-F and (His)₆-tagged Pcthis-R using pPs619C1437Pct540 as template. The amplified DNA fragment was digested with *Nde*I and *Bam*HI and ligated with *Nde*I-*Bam*HI-digested pET22b(+) plasmid. To construct pET22b-his4CL, the *S. coelicolor* 4-coumarate:CoA ligase gene (*4CL*) was amplified with the primers 4CLhis-F and 4CLhis-R using the genomic DNA of *S. coelicolor* as a template. The PCR product was digested with *Eco*RI and *Sbf*I, and ligated with *Eco*RI-*Sbf*I-digested pET22b(+). To construct pET22b-his4CL(A294G), the first 885 bp DNA fragment was amplified with the primers 4CLhis-F and 4CLmut-R containing a single mutated base (G881C). The second 720 bp DNA fragment was amplified with the primers 4CLmut-F containing a single mutated base (C881G) and 4CLhis-R. Then the complete 1,587 bp DNA fragment was amplified with the primers 4CLhis-F and 4CLhis-R by overlapping PCR using the mixed two fragments as a template. To make pET22b-hisFldA, the *C. botulinum* A str. ATCC 3502 cinnamoyl-CoA:phenyllactate CoA-transferase gene (*fldA*) was used. For better expression of the *fldA* gene in recombinant *E. coli*, its codon usage was further optimized to *E. coli* (synthesized at GenScript, Piscataway, NJ, and cloned into a vector to make plasmid pUC57-FldAopt) and *E. coli* codon-optimized *fldA* gene was amplified with the primers FldAhis-F and FldAhis-R using pUC57-FldAopt as a template. The PCR product was digested with *Nde*I and *Hin*dIII, and ligated with *Nde*I-*Hin*dIII-digested pET22b(+) plasmid.

To construct pKM212-AroG$^{fbr}$ for the expression of the feedback inhibition resistant, DAHP synthase gene (*aroG*) mutant was amplified by PCR with the primers AroG-F and AroG-R using the plasmid pTyr-a[27] as a template. The PCR product was digested with *Eco*RI and *Hin*dIII, and ligated with *Eco*RI-*Hin*dIII-digested pKM212-MCS[12]. The pKM212-AroG$^{fbr}$PheA$^{fbr}$ plasmid was constructed as follows. First, the 991 bp DNA fragment was amplified by PCR from the genomic DNA of *E. coli* using primers PheA-F and PheAmut-R, which contains a single mutated base (T976G). Second, 200 bp DNA fragment was amplified from the genomic DNA of *E. coli* using the primers PheAmut-F containing a single mutated base (A976C) and PheA-R. Then, complete 1,161 bp DNA fragment of the *pheA*$^{fbr}$ gene was amplified with the primers PheA-F and PheA-R by overlapping PCR using mixed two fragments as a template. The PCR product was digested with *Hin*dIII and ligated with *Hin*dIII-digested pKM212-AroG$^{fbr}$. To construct pKM212-AroG$^{fbr}$PAL, the codon usage of *Streptomyces maritimus PAL* gene was further optimized to *E. coli* (synthesized at GenScript and cloned into a vector to make plasmid pUC57-PALopt). The *E. coli* codon-optimized PAL was amplified by PCR with the primers PAL-Hin-F and PAL-Hin-R using pUC57-PALopt. The PCR product was digested with *Hin*dIII and ligated with *Hin*dIII-digested pKM212-AroG$^{fbr}$. For the construction of pACYC-FldH, the *C. botulinum* A str. ATCC 3502 D-phenyllactate dehydrogenase gene (*fldH*) was used. The codon usage of the *fldH* gene was optimized to *E. coli* (synthesized at GenScript and cloned into a vector to make plasmid pUC57-FldHopt) and the *E. coli* codon optimized *fldH* gene was amplified with the primers FldH-F and FldH-R using the pUC57-FldHopt as a template. The PCR product was digested with *Bam*HI and *Hin*dIII, and ligated with *Bam*HI-*Hin*dIII-digested pTrc99A to make pTrc-FldH. Then, the *fldH* gene attached to the *trc* promoter and *rrnB* terminator was amplified by PCR with the primers Trc-F and Ter-R using pTrc-FldH as a template. The PCR product was digested with *Xho*I and *Sac*I, and ligated with *Xho*I-*Sac*I-digested pACYC184KS to construct pACYC-FldH.

To construct pACYC-4CL(A294G), the mutant *4CL* having amino acid change (A294G) was amplified by PCR with the primers Trc-F and Ter-R using the

plasmid pET22b-his4CL(A294G) as a template. The PCR product was digested with XhoI and SacI, and ligated with XhoI–SacI-digested pACYC184KS to construct pACYC-4CL(A294G). pACYC-4CL(A294G)FldA was constructed by cloning the fldA gene amplified by PCR using the primers FldA-F and FldA-R containing C-terminal (His)₆-tag from pET22b-hisFldA as a template into pACYC-4CL(A294G) at SbfI site. To make pACYC-4CL(A294G)FldAH, the fldH gene was amplified by PCR with the primers FldH-sbF and FldH-hiR using the plasmid pACYC-FldH as a template. The PCR product was digested with SbfI and HindIII sites, and ligated with SbfI–HindIII-digested pACYC-4CL(A294G)FldA to construct pACYC-4CL(A294G)FldAH.

To construct pET22b-hisHadA, the C. difficile HadA gene (hadA; AAV40822.1) fused with C-terminal (His)₆-tag was amplified by PCR using the primers HadA-hisF and HadA-hisR using the genomic DNA of C. difficile as a template. The PCR product was digested with NdeI and NotI, and ligated with NdeI–NotI-digested pET22b(+) plasmid. pPs619C1437-HadA was constructed by replacing the pct540 gene in pPs619C1437Pct540 with the C. difficile hadA gene that was amplified by PCR with primers HadA-sbF and HadA-ndR using the genomic DNA of C. difficile as a template. The PCR product was digested with SbfI and NdeI, and ligated with SbfI–NdeI-digested DNA fragment of pPs619C1437Pct540.

For the construction of pKM212-GPE_PhaAB, the R. eutropha PHA biosynthesis operon promoter, the β-ketothiolase (phaA), and acetoacetyl-CoA reductase (phaB) genes were amplified by PCR with the primers PhaAB-BamF and PhaAB-sbR using plasmid pCnAB[7] as a template. The PCR product was digested with BamHI and SbfI, and ligated with BamHI–SbfI-digested pKM212-AroG^fbrPheA^fbr. To construct plasmids having different expression level of the phaAB genes under synthetic Anderson promoters (http://parts.igem.org/), the phaAB operon linked with BBa_J23100 was amplified by primer 100-Kpn-F and PhaB-Bam-R using the genomic DNA of R. eutropha as a template. The PCR product was digested with KpnI and BamHI, and ligated with KpnI–BamHI-digested pKM212-AroG^fbrPheA^fbr. Then, BBa_J23100 region was replaced with BBa_J23105, BBa_J23114, BBa_J23109, and BBa_J23103 promoters, having relative promoter strengths of 1, 0.24, 0.10, 0.04, and 0.01, respectively (http://parts.igem.org/) through the same manner using corresponding primers and restriction enzymes as mentioned above. Deletion of genes in the chromosome of E. coli XL1-Blue was performed by the one step inactivation method[28]. The primers used are listed in Supplementary Table 4. For replacements of ldhA gene and its native promoter with the fldH gene and strong trc promoter in the chromosome of E. coli XB201TBA, plasmid pMtrcFldH, bearing a trc promoter and fldH gene downstream of the lox66-cat-lox71 cassette, was used.

**In silico flux response analysis**. The genome-scale metabolic model of E. coli iJO1366[29], comprising 2,251 metabolic reactions and 1,135 metabolites, was used for in silico flux response analysis to examine the effects of central and aromatic amino acid biosynthesis reactions on the D-phenyllactic acid production[30]. In order to reflect the genotype of XB201T strain in the iJO1366, heterologous metabolic reactions biosynthesizing D-phenyllactate (fldH gene) were additionally introduced to the model and, and flux values of native E. coli reactions were constrained to zero for the genes knocked out in the XB201T strain. For the implementation of in silico flux response analysis, the flux values of each central and aromatic amino acid biosynthesis reaction were constrained from their minimum values to the maximum values. D-phenyllactic acid production rate (D-phenyllactate dehydrogenase) was subsequently maximized as an objective function at each constraint point where flux values of each central and aromatic amino acid biosynthesis reaction were fixed. The relationships between D-phenyllactic acid production rate and fluxes of each central and aromatic amino acid biosynthesis reaction were categorized into six types based on the flux patterns obtained from the in silico flux response analysis. Reactions that were predicted to be negatively correlated with D-phenyllactic acid production rate were identified as potential knockout candidates. During the simulation, the glucose uptake rate was set at 10 mmol per gram of dry cell weight per hour. All simulations were performed in Python environment with Gurobi Optimizer 6.0 and GurobiPy package (Gurobi Optimization, Inc., Houston, TX). Reading, writing, and manipulation of the COBRA-compliant SBML files were implemented using COBRApy[31].

**Culture condition**. Luria–Bertani (LB) medium (containing per liter: 10 g tryptone, 5 g yeast extract and 10 g NaCl) was used for cultivation of E. coli for DNA manipulations. Recombinant E. coli XL1-Blue strains were cultured at 30 °C in a chemically defined MR medium at 200 r.p.m. in a rotary shaker for the production of PHAs. MR medium (pH 7.0) contains (per liter): 6.67 g KH₂PO₄, 4 g (NH₄)₂HPO₄, 0.8 g MgSO₄·7H₂O, 0.8 g citric acid, and 5 ml trace metal solution. The trace metal solution contains (per liter of 0.5 M HCl): 10 g FeSO₄·7H₂O, 2 g CaCl₂, 2.2 g ZnSO₄·7H₂O, 0.5 g MnSO₄·4H₂O, 1 g CuSO₄·5H₂O, 0.1 g (NH₄)₆Mo₇O₂₄·4H₂O, and 0.02 g Na₂B₄O₇·10H₂O. Glucose, MgSO₄·7H₂O, sodium 3HB (Acros Organics), D-phenyllactic acid, D-mandelic acid, and D-3-hydroxy-3-phenylpropionic acid were sterilized separately. When the tyrB and aspC deletion strains were cultivated, 0.2 g l⁻¹ of L-tyrosine, 0.2 g l⁻¹ of L-phenylalanine, and 3 g l⁻¹ of L-aspartic acid were added to the culture medium. Sodium 3HB (1 g l⁻¹) was added to the culture medium when it was needed. Ampicillin (Ap, 50 µg ml⁻¹), kanamycin (Km, 30 µg ml⁻¹), and chloramphenicol (Cm, 34 µg ml⁻¹) were added to the medium according to the resistant marker of employed plasmids.

Fed-batch cultures were carried out in a 6.6 l Bioreactor (Bioflow 3000; New Brunswick Scientific Co., Edison, NJ) containing 2 l of MR medium supplemented with 20 g l⁻¹ of glucose and 1 g l⁻¹ of sodium 3HB. The dissolved oxygen concentration was maintained above 40% of air saturation by automatically supplying air with the rate of 2 l min⁻¹ and by automatically controlling the agitation speed from 200 to 1,000 r.p.m. The culture pH was controlled at 7.0 by 28% (v/v) ammonia solution. The feeding solution contains (per liter): 700 g glucose, 15 g MgSO₄·7H₂O, and 250 mg thiamine. The fed-batch fermentation was performed by pH-stat strategy and also by pulsed-feeding strategy. For the pH-stat fed-batch cultures, the pH of fermentation broth was maintained at pH 7.0 and the feeding solution was automatically fed when the pH rose above 7.02 as a set point. For pulsed-feeding strategy, feeding solution was added when the glucose concentration in the fermentation broth fell below 1 g l⁻¹ to increase the glucose concentration in the culture broth to 20 (in the case of 3HB supplemented fed-batch cultures) or 10 g l⁻¹ (without 3HB supplemented fed-batch cultures).

**Pct and HadA enzyme assay**. To assay the activity of Pct540 and HadA, E. coli BL21 (DE3) harboring pET22b-hisPct540 or pET22b-hisHadA was cultured in LB medium at 30 °C and each enzyme was purified by using Talon metal affinity resin (Clontech, Mountain View, CA). The activities of Pct540 and HadA were measured as follows: the reaction was carried out for 30 min at 30 °C in 50 mM phosphate buffer (pH 7.5) containing 0.1 mM acetyl-CoA and 10 mM substrates by adding 15 µg of purified enzyme. After the above reaction, 1 mM oxaloacetate, 5 µg of citrate synthase, and 0.5 mM 5,5′-dithiobis-(2-nitrobenzoic acid) were added. Then, amount of released free CoA was analysed by measuring the absorbance at 412 nm[32]. One unit of enzyme activity was defined as the formation of 1 µmol of corresponding CoA derivative from the substrate per minute. The specific activity was described as unit mg⁻¹ protein.

**4CL enzyme assay**. The activity of 4CL was determined by the spectrophotometric assay as follows: the changes in absorbance during cinnamoyl-CoA formation were monitored at wavelengths of 311 nm. The reaction mixture contained 400 mM Tris-HCl (pH 7.8), 5 mM ATP, 5 mM MgCl₂, 0.3 mM CoA, 0.5 mM cinnamic acid, and 50 µg of 4CL in total volume of 1 ml. The reaction was started by the adding 4CL enzyme at 30 °C for 5 min. The change in absorbance was measured by successive scanning of the wavelength with a spectrophotometer[18,33].

**Cinnamoyl-CoA preparation and FldA enzyme assay**. The cinnamoyl-CoA for FldA reaction was prepared as follows: the OASIS HLB SPE cartridge (Waters, Milford, MA) was first conditioned with 3 ml of methanol followed by addition of 3 ml of 0.15% trichloroacetic acid (TCA) solution. The 4CL enzyme assay mixture was applied to the cartridge followed by addition of 2 ml of 0.15% TCA solution. The attached CoA-thioesters were eluted with 1 ml of methanol–ammonium hydroxide (99: 1, v/v) and then the sample was further evaporated to dryness at room temperature by vacuum centrifugation at 15,493×g. The sample was dissolved with 50 mM phosphate buffer (pH 7.5). The reaction was carried out for 1 h at 30 °C in 50 mM phosphate buffer (pH 7.5) containing cinnamoyl-CoA and 10 mM substrates by adding 10 µg of purified FldA. After reaction, the consumption of cinnamoyl-CoA was measured by the spectrophotometric assay by measuring the absorbance at 311 nm.

**Analysis of aliphatic and aromatic acyl-CoAs**. The formation of CoA-thioesters was confirmed by using high-performance liquid chromatography (HPLC)-mass spectrometry (MS). For the analysis of aliphatic and aromatic acyl-CoAs, samples were prepared by method described above in cinnamoyl-CoA preparation. The prepared samples were dissolved with 1 ml of water for analysis and were subjected to HPLC (1100 series HPLC, Agilent Technologies, Palo Alto, CA) equipped with MS (LC/MSD VL, Agilent) and Eclipse XDB-C18 column (5 µm, 4.6 × 150 mm, Agilent).

**Cell growth and metabolite analysis**. Glucose concentration was measured using glucose analyser (model 2700 STAT, Yellow Spring Instrument, OH). Cell growth was monitored by measuring absorbance at 600 nm (OD₆₀₀) using an Ultrospec 3000 spectrophotometer (Amersham Biosciences, Uppsala, Sweden). The metabolites (acetate, formate, lactate, succinate, and pyruvate) were analysed by HPLC (1515 isocratic HPLC pump, Waters) equipped with refractive index detector (2414, Waters) and MetaCarb 87 H column (Agilent). Elution was performed with 0.01 N H₂SO₄ at a flow rate of 0.5 ml min⁻¹ at 25 °C. D-Phenyllactic acid, D-4-hydroxyphenyllactic acid, L-tyrosine, and L-phenylalanine were analysed by HPLC (1100 series HPLC, Agilent Technologies) equipped with UV/VIS (G1315B, Agilent), refractive index (Shodex RI-71; Tokyo, Japan) detector and a ZorbaxSB-C18 column (150 mm × 4.6 mm, 5 µm, Agilent). Elution was performed with mobile phase A (Water-0.1% trifluoroacetic acid) and mobile phase B (Acetonitrile). A linear gradient scheme started from 10% of mobile phase A and the fraction of mobile phase A was increased up to 50%. The flow rate was 1 ml min⁻¹ and analytes were detected at 220 nm.

**Polymer analysis**. The polymer contents and monomer compositions of synthesized PHAs were determined by gas chromatography (GC) or GC-MS (Supplementary Methods)[34]. The collected cells were washed three times with distilled water and then lyophilized for 24 h. The PHAs in lyophilized cells were converted into corresponding hydroxymethyl esters by acid-catalyzed methanolysis[33]. The resulting methyl esters were analysed by GC (Agilent 6890 N, Agilent) equipped with Agilent 7683 automatic injector, flame ionization detector, and a fused silica capillary column (ATTM-Wax, 30 m, ID 0.53 mm, film thickness 1.20 μm, All-tech). Polymers were extracted from the cells by chloroform extraction method[35,36]. Polymer content is defined as the weight percentage of polymer concentration to dry cell concentration (wt% of dry cell weight). The structure, molecular weights, and thermal properties of the polymers are determined by nuclear magnetic resonance spectroscopy, gel permeation chromatography, and differential scanning calorimetry, respectively (Supplementary Methods)[8,9].

**Transmission electron microscopy analysis**. To confirm intracellular aromatic PHA synthesis in recombinant *E. coli*, transmission electron microscopy (TEM) analysis was performed. After 72 h of cultivation, 5 ml of cells were collected and washed twice with phosphate-buffered saline (pH 7.0). Cells were fixed by adding 2.5% (v/v) glutaraldehyde in phosphate-buffered saline (pH 7.0) and 1% (v/v) osmium tetroxide. Then, the samples were dehydrated and embedded in Epon 812. The ultra-thin section was carried out by Ultra-Microtome (Leica Ultracut UCT, Leica, Austria). The section was stained with 2% uranyl acetate and lead citrate. TEM analysis was performed using Bio-TEM (Tecnai G2 Spirit, FEI, Hilsboro, OR) at the Korea Basic Science institute.

**Grazing incidence X-ray scattering measurements**. The samples for grazing incidence X-ray scattering (GIXS) measurements were prepared by spin coating of copolymer solutions in chloroform (2 wt%) on silicon substrate. GIXS measurements were performed on beamline 3 C in the Pohang Accelerator Laboratory (South Korea). X-rays with energy of 11.075 keV were used and the off-specular scattering was recorded using a MAR-CCD area detector and sample to detector distance was maintained to be 30 cm. The incidence angle of the X-ray beam (~ 0.12°) was used to allow for complete penetration of X-ray beam into the polymer film.

**Dual cantilever beam test**. The dual cantilever beam (DCB) specimens were prepared with the geometry of glass/aromatic PHAs/Au/epoxy/glass. A series of PHAs was dissolved in chloroform at the concentration of 2 wt% and then spin-coated on a bare glass substrate. After sufficient drying of the residual chloroform in a vacuum oven, a 50 nm-thick Au layer is deposited on the top of the polymer films by thermal evaporation. And then, the specimen was cut into 8 mm × 25.4 mm-long glass beam using Dicing Machine (DAD3350, DISCO Co., Japan) in order to prevent edge defects in the glass substrates. Finally, a dummy glass substrate coated by B-stage epoxy (Epo-Tek M10-D; Epoxy Technology) with a 1 μm thickness was attached to complete the preparation of the DCB specimens. Two aluminum taps were attached at the specimen to apply loading and delaminate the DCB specimen. The cohesion energy was measured using a high-precision micromechanical test system (Delaminator Adhesion Test System, DTS Company, Menlo Park, CA). Multiple loading/crack-growth/unloading cycles were performed under a constant displacement rate of 0.5 μm s$^{-1}$, whereas the applied load was continuously recorded as a function of the displacement. The cohesion energy ($G_c$) was calculated as the critical value of the strain energy release rate[37,38].

**Statistical analysis**. All experiments were performed at least three times. Means were compared using a two-sided Student's *t*-test. $P < 0.05$ was considered significant. The investigators were blinded to the group allocation by randomly selecting single colonies multiple times. No statistical methods were used to predetermine sample size.

**Data availability**. The data supporting the findings of this study are available from the authors upon request.

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

## Acknowledgements

This work was supported by the Intelligent Synthetic Biology Center through the Global Frontier Project (2011-0031963) and also by the Technology Development Program to Solve Climate Changes on Systems Metabolic Engineering for Biorefineries (NRF-2012M1A2A2026556 and NRF-2012M1A2A2026557) from the Ministry of Science and ICT through the National Research Foundation of Korea.

## Author contributions

S.Y.L. conceived the project. S.Y.L., J.E.Y., and S.J.P. generated ideas and designed research. J.E.Y. and S.J.P. performed research and analytical experiments. W.J.K. performed in silico metabolic simulation. J.E.Y., S.J.P., S.Y.L., H.J.K., B.J.K., J.S., and H.L. analysed data. J.E.Y., S.J.P., and S.Y.L. wrote the paper and all authors approved the final manuscript.

## Additional information

**Competing interests:** S.Y.L., J.E.Y., and S.J.P. declare that they have competing financial interest as strains, enzymes, and genes described in this paper are patented and are of commercial interest. The remaining authors declare no competing financial interests.

