## [Peer Review File · Nature Communications]

Reviewers' comments:

Reviewer #1 (Remarks to the Author):

Aromatic polyesters are widely used materials. For example, polyethylene terephthalate (PET) has broad applications ranging from Coca bottle to clothing fiber. Based their previous work, the authors report one-step fermentative production of aromatic polyesters from glucose by metabolically engineered *Escherichia coli* strains expressing coenzyme A transferase and polyhydroxyalkanoate synthase. Various aromatic polyesters containing phenyllactate, D-mandelate and D-3-hydroxy-3-phenylpropionate are produced around 1 g/L. This manuscript is more suitable for a specialized Journal such as Metabolic Engineering.

Comments:

1. The strategy is not completely novel. It is extended from the authors' previous work (Choi, S. Y. et al. One-step fermentative production of poly(lactate-co-glycolate) from carbohydrates in *Escherichia coli*. *Nat. Biotechnol.* 34, 435-440 (2016)). Here the precursors are changed to aromatic metabolites.
2. For material application, performance and cost are the major considerations. Here the titer of the polymer is less than 1 g/L. Considering the theoretical yield and titer, it is very hard to envision its competition against polymers like PET, which has a price of less than \$2/kg. Such aromatic polymers can also come from cheaper feedstocks such as lignin.
3. The authors contribute applaudable efforts to characterize the material properties. However, the conclusion that "These results suggest that aromatic polyesters produced by one-step direct fermentation of engineered *E. coli* strains can be used to replace those aromatic polyesters currently produced from petroleum" is confusing. It would be helpful if the authors compare common properties, such as T_m , T_g , and tensile strength, between these two types of polymers. I only see low T_g here and the material property looks inferior to that of PET. I am interested in knowing what polymers and what particular applications can be better replaced by this new polymer.

Reviewer #2 (Remarks to the Author):

The manuscript comprises the use of two CoA-transferases (FldA and HadA) involved in

clostridial amino acid fermentations to engineer *E. coli* for the production of polyesters containing aromatic residues. The experiments are well performed and described. The statistical analysis is appropriate. I have only a few minor comments which may improve the manuscript:

1. Abstract, line 4: Only genes are expressed, see line 9.
2. Abstract, line 9: Define fldH. I am not sure whether most readers are familiar with the other gene abbreviations: aroG, pheA, tyrR, pflB, poxB, adhE, frdB, tyrB and aspC.
3. Page 3, line 21: The correct name of Pct is propionate CoA-transferase.
4. Page 5, line 17: 99% sequence identity!
5. Page 7, line 18: Glucose (20 g L⁻¹).
6. Page 11, line 4: define PET!
7. Page 11, line 6: molecular mass (Da).
8. Page 11, line 16: define Tg!
9. Page 15, line 21: 2-hydroxyisocaproate CoA-transferase
10. Page 16, line 24: explain how the flux analysis was performed.
11. The figures have no numbers.
12. Fig. 3. The structures most likely should represent the methylated form of poly(3HB-co-D-phenyllactate), but on C1 the methyl group should be replaced by methoxy group.
13. Fig. S1. The negative results should be combined with those of Fig. S7.
14. Figs. S2 and S3: Explain that the small peak at m/z + 22 represents the monosodium form.
15. Fig. S5a. The structure represents 1-phenyl-2-methoxy-3-oxobutane but not methyl phenyllactate.
16. Fig. S5b. In the 4-hydroxyphenyllactate moiety on C1 the methyl group should be replaced by methoxy group.

Wolfgang Buckel

Reviewer #3 (Remarks to the Author):

Authors reported a one-step fermentative production of aromatic polyesters from glucose by metabolically engineered *Escherichia coli* strains expressing a newly identified coenzyme A (CoA) transferase and an evolved polyhydroxyalkanoate (PHA) synthase. Isocaprenoyl-CoA:2-hydroxyisocaproate CoA-transferase (HadA) capable of generating various aromatic hydroxyacyl-CoAs using acetyl-CoA as a CoA-donor. This is something really new and innovative. The paper can be accepted after following minor issues are properly addressed:

1. Page 6, lines 2-3 "4CL and FldA can potentially be used for phenyllactyl-CoA generation and consequently aromatic polyester production". The 4CL and FldA are two key enzymes for phenyllactyl-CoA generation, authors are suggested to provide a little bit more info on these two

enzymes even though they have supplementary info (most readers do not usually go to see the supplementary info).

2. The newly synthesized aromatic PHAs look interesting. Are they random or block copolymers? The way authors described their structures, they are block copolymers, yet authors said that they are random copolymer. The NMR description is not clear to me. Please clarify them based on the NMR, better with a DSC graph.

3. The aromatic PHA containing 3HB is new in the area. The authors should provide some characterizations on molecular weights, thermal property et al although mechanical property is difficult at this stage due to the small amount of materials available.

4. Page 9, lines 23-25 "poly(55.2 mol% 3HB-co-43.0 mol% D-phenyllactate-co-1.8 mol% D-mandelate)". This is also a new PHA. However, I would assume this is a very sticky material without a clear application area. Please elaborate.

5. The PHA content in cell dry weight (CDW) is not high in this study, maybe the mutated PhaC has a low activity. Could authors find a way to increase PHB content? for example, by using a stronger promoter for phaCAB? or by optimizing RBS?

6. The novelty of this paper is "the development of a bacterial platform system that allows production of various aromatic polyesters by one-step fermentation from glucose". Could authors make aromatic monomers from glucose for polymerization? This could generate better polymer properties. Could authors discuss this possibility?

Responses to the Editor's and Reviewers' comments

Manuscript number: NCOMMS-17-16203-T

***Our responses are shown in BLUE, while those also included in the revised manuscript are shown in RED.

[Editor's comments]

Your manuscript entitled "Microbial production of aromatic polyesters from glucose" has now been seen by 3 referees, whose comments are appended below. You will see from their comments copied below that while they find your work of considerable potential interest, they have raised quite substantial concerns that must be addressed. In light of these comments, we cannot accept the manuscript for publication, but would be interested in considering a revised version that addresses these serious concerns.

We hope you will find the referees' comments useful as you decide how to proceed. Should further experimental data or analysis allow you to address these criticisms, we would be happy to look at a substantially revised manuscript. Specifically, we would like you to seek strategies to further increase titer yield including those suggested by R#3. Please bear in mind that we will be reluctant to approach the referees again in the absence of major revisions. If the revision process takes significantly longer than three months, we will be happy to reconsider your paper at a later date, as long as nothing similar has been accepted for publication at Nature Communications or published elsewhere in the meantime.

[RESPONSE] Thank you very much for your kind effort to improve our manuscript. We performed additional experiments according to your and reviewers' comments to increase the titer. In order to include new results on enhanced production of aromatic polyesters from glucose without 3HB supplementation, we performed the following additional experiments. First, pH-stat fed-batch culture of XB201TBAL strain expressing AroG^{fbr}, PheA^{fbr}, FldH, HadA, PhaC1437 and PhaAB under BBa_J23114 promoter was performed in a medium containing glucose without 3HB feeding, which produced 2.5 g l⁻¹ of poly(67.6 mol% 3HB-co-32.4 mol% D-phenyllactate) with a polymer content of 43.8 wt%. Although this result is already higher than that we reported in the original manuscript with 3HB feeding, we decided to further enhance aromatic polyester production. For this, we engineered *E. coli* strain thorough integrating the key enzyme (FldH) under stronger promoter. Furthermore, we found that pulsed-feeding fed-batch culture was better than pH-stat fed-batch culture. We were able

to produce 13.9 g l⁻¹ of poly(61.9 mol% 3HB-co-38.1 mol% D-phenyllactate) with a polymer content of 55.0 wt% from only glucose by fed-batch fermentation (Supplementary Fig. 9). These values are significantly higher than those we reported in the original manuscript (see detailed response below). These results demonstrate that aromatic PHAs could be successfully produced to a reasonably high concentration by fed-batch culture of the engineered strain (XB201TBAF strain expressing AroG^{fbr}, PheA^{fbr}, HadA, PhaC1437 and PhaAB under BBa_J23114 promoter). We want to thank the editor and the reviewers very much for the great comments which allowed further improvement of our manuscript.

Detailed responses to the reviewers' comments are shown below.

[Reviewers' comments]

Reviewer #1 (Remarks to the Author):

Aromatic polyesters are widely used materials. For example, polyethylene terephthalate (PET) has broad applications ranging from Coca bottle to clothing fiber. Based their previous work, the authors report one-step fermentative production of aromatic polyesters from glucose by metabolically engineered *Escherichia coli* strains expressing coenzyme A transferase and polyhydroxyalkanoate synthase. Various aromatic polyesters containing phenyllactate, D-mandelate and D-3-hydroxy-3-phenylpropionate are produced around 1 g/L. This manuscript is more suitable for a specialized Journal such as *Metabolic Engineering*.

[RESPONSE] Thank you for your effort in reviewing our manuscript. We believe that this paper is suitable for *Nature Communications* as it has not been possible to produce aromatic polyesters by one-step microbial fermentation, which we show for the first time in this paper.

Comments:

1. The strategy is not completely novel. It is extended from the authors' previous work (Choi, S. Y. et al. One-step fermentative production of poly(lactate-co-glycolate) from carbohydrates in *Escherichia coli*. *Nat. Biotechnol.* 34, 435-440 (2016)). Here the precursors are changed to aromatic metabolites.

[RESPONSE] Changing precursors to aromatic metabolites itself is not trivial and could not be demonstrated over the last 30 years in microbial polyester synthesis. Furthermore, one-step fermentative production of aromatic polyesters is not possible using the previous systems for PLGA production we reported in "*Nature Biotechnology*" by simply changing the precursors to aromatic ones. As we clearly described in our manuscript, the following

strategies had to be developed and employed in an integrated manner to accomplish one-step fermentative production of aromatic polyesters from glucose. The first key strategy was to identify novel CoA transferases that can efficiently activate phenylalkanoates into their corresponding CoA derivatives. The second strategy was to metabolically engineer *E. coli* to produce phenylalkanoates from glucose. The third strategy was to integrate metabolic pathways for producing aromatic precursors with the polymerization system. The fourth strategy was to produce various aromatic polyesters having different monomer fractions through modulating the enzyme expression levels. All these strategies should be successfully combined for the production of aromatic polyesters by one-step fermentation of engineered *E. coli* strains. Furthermore, we developed strategies for the production of other aromatic polyesters containing D-mandelate and D-3-hydroxy-3-phenylpropionate, which strongly suggests that the systems metabolic engineering strategies developed in this study should be useful for the production of novel aromatic polyesters by one-step fermentation.

2. For material application, performance and cost are the major considerations. Here the titer of the polymer is less than 1 g/L. Considering the theoretical yield and titer, it is very hard to envision its competition against polymers like PET, which has a price of less than \$2/kg. Such aromatic polymers can also come from cheaper feedstocks such as lignin.

[RESPONSE] Development of the overall integrated strategies for the one-step fermentative production of aromatic polyesters for the first time was already challenging. We did not intend to perform optimization studies for the increased production of aromatic polyesters in this study; further improvement of titer, yield and productivity of aromatic polyesters are targets of our next study. Although we do not think that we can include all the results (e.g., increased aromatic polyester production in this case) in one paper, we tried our best to show that further engineering will allow enhanced production of aromatic polyesters. As the editor and reviewers suggested, however, we conducted additional experiments (further strain engineering and fed-batch cultures) during the revision process. Through additional experiments, we were able to show that one-step direct fermentative production of aromatic polyesters is indeed possible; we were able to increase the concentration of aromatic polyester, poly(61.9 mol% 3HB-co-38.1 mol% D-phenyllactate) to 13.9 g l⁻¹ by performing re-design and optimization of metabolic pathways of *E. coli* host strains and developing fed-batch fermentation strategies. We added the following new results in page 11. Also, we described fermentation strategies in Methods section.

“Next, the pH-stat fed-batch culture of *E. coli* XB201TBAL strain expressing AroG^{fbr}, PheA^{fbr}, FldH, HadA, PhaC1437 and PhaAB under BBa_J23114 promoter was performed in a medium containing glucose without 3HB feeding. In 96 h, 2.5 g l⁻¹ of poly(67.6 mol% 3HB-co-32.4 mol% D-phenyllactate) was produced with a polymer content of 43.8 wt%, which demonstrated that aromatic polyester can be produced by one-step fermentation of engineered *E. coli* from glucose (Supplementary Fig. 9). To further enhance production of aromatic PHAs, gene expression system was optimized by replacing the *ldhA* gene with the *fldH* gene in the chromosome of *E. coli* XB201TBA. Also, the native promoter of the *ldhA* gene was replaced with the strong *trc* promoter to increase *fldH* expression. Furthermore, we employed pulsed-feeding method, which gave better polymer production (see Methods; data not shown). The resulting *E. coli* XB201TBAF strain expressing AroG^{fbr}, PheA^{fbr}, HadA, PhaC1437 and PhaAB under BBa_J23114 promoter allowed production of 13.9 g l⁻¹ of poly(61.9 mol% 3HB-co-38.1 mol% D-phenyllactate) with a polymer content of 55.0 wt% from glucose by fed-batch fermentation (Supplementary Fig. 9). This titer (13.9 g l⁻¹) obtained is 5.56 fold higher than that (2.5 g l⁻¹) obtained with the *E. coli* XB201TBAL strain expressing AroG^{fbr}, PheA^{fbr}, FldH, HadA, PhaC1437 and PhaAB under BBa_J23114 promoter, and is also much higher than that (less than 1 g l⁻¹) obtained by fed-batch culture of the *E. coli* XB201TBAL strain expressing AroG^{fbr}, PheA^{fbr}, FldH, HadA and PhaC1437 in a medium supplemented with glucose and sodium 3HB. These results demonstrate that aromatic PHAs could be successfully produced to a reasonably high concentration by fed-batch culture of the engineered strain (XB201TBAF strain expressing AroG^{fbr}, PheA^{fbr}, HadA, PhaC1437 and PhaAB under BBa_J23114 promoter). Although we provided proof-of-concept fermentation results here, it is expected that further optimization of cultivation condition will allow more efficient production of aromatic polyesters by one-step fermentation from glucose.”

Newly added Supplementary Figure 9d-9g are shown below:

3. The authors contribute applaudable efforts to characterize the material properties. However, the conclusion that “These results suggest that aromatic polyesters produced by one-step direct fermentation of engineered *E. coli* strains can be used to replace those aromatic polyesters currently produced from petroleum” is confusing. It would be helpful if the authors compare common properties, such as T_m , T_g , and tensile strength, between these two types of polymers. I only see low T_g here and the material property looks inferior to that of PET. I am interested in knowing what polymers and what particular applications can be better replaced by this new polymer.

[RESPONSE] Thank you. We have already provided the polymer properties in our original manuscript. The molecular weight ranged from 3.6 to 24.9 kDa, which is suitable for making fiber grade PET. In addition, our newly synthesized aromatic polyester shows reduced crystallinity (Fig. 4), increased T_g (33.47 °C) and G_c ($0.90 \pm 0.12 \text{ J m}^{-2}$) values compared with

those of poly(3HB), which is highly crystalline having T_g of $-3.13\text{ }^\circ\text{C}$ and G_c of $0.28 \pm 0.11\text{ J m}^{-2}$. Thus, the aromatic polyesters produced in this study can be a promising substitute for currently used petro-based aromatic polyesters. Although more polymer characterization study is needed, for now, our polyester can be used to replace fiber grade PET of petrochemical origin through further improvement. Furthermore, it is well known that mandelate, one of our proof-of-concept monomer, has the same properties as polystyrene when chemically polymerized.

Reviewer #2 (Remarks to the Author):

The manuscript comprises the use of two CoA-transferases (FIdA and HadA) involved in clostridial amino acid fermentations to engineer *E. coli* for the production of polyesters containing aromatic residues. The experiments are well performed and described. The statistical analysis is appropriate. I have only a few minor comments which may improve the manuscript:

[RESPONSE] Thank you very much for appreciating the importance of our work.

1. Abstract, line 4: Only genes are expressed, see line 9.

[RESPONSE] Thank you. Please see our response to #2 together.

2. Abstract, line 9: Define fldH. I am not sure whether most readers are familiar with the other gene abbreviations: aroG, pheA, tyrR, pflB, poxB, adhE, frdB, tyrB and aspC.

[RESPONSE] Thank you. We could not describe the full names of the genes/enzymes due to the restriction in the word count. However, we think that it will be good to explain the less familiar gene such as *fldH*. Thus, we added “(D-phenyllactate dehydrogenase)” for *fldH*. The full names for all the genes used are described in the main text.

3. Page 3, line 21: The correct name of Pct is propionate CoA-transferase.

[RESPONSE] Thank you. Actually, both names are correct. Since we and others have used propionyl-CoA transferase before, we want to stick to this name.

4. Page 5, line 17: 99% sequence identity!

[RESPONSE] Thank you for the comment. We made it clear as you suggested.

“99.0% of amino acid sequence identity”

5. Page 7, line 18: Glucose (20 g L⁻¹).

[RESPONSE] We want to stick to 20 g L⁻¹ glucose rather than glucose (20 g l⁻¹) as this is the style we used throughout the manuscript.

6. Page 11, line 4: define PET!

[RESPONSE] We already defined it on Page 4, line 10. “poly(ethylene terephthalate) [PET]”

7. Page 11, line 6: molecular mass (Da).

[RESPONSE] Thank you very much. In the polymer field, it is standard to use a number-averaged molecular weight as kDa or Da, although more accurately, kDa or Da represents molecular mass as you suggested. To make this clear, we added “molecular mass” in the parenthesis.

“have a number-average molecular weight (M_n ; **molecular mass**) of 15 to 20 kDa and 24 to 36 kDa, respectively.”

8. Page 11, line 16: define T_g!

[RESPONSE] Thank you. We defined T_g.

“...high **glass transition temperature (T_g)** of the polymer.”

9. Page 15, line 21: 2-hydroxyisocaproate CoA-transferase

[RESPONSE] Thank you. Corrected.

“... **2-hydroxyisocaproate CoA-transferase** gene (*hadA*)”

10. Page 16, line 24: explain how the flux analysis was performed.

[RESPONSE] Thank you. We modified the sentence in the Methods section to clarify as follows in page 17 line 5;

“The genome-scale **metabolic** model of *E. coli* iJO1366³⁰, comprising 2251 metabolic **reactions and 1135 metabolites**, was used for *in silico* flux response analysis to examine the effects of central and aromatic amino acid biosynthesis reactions on the D-phenyllactic acid production³¹. **In order to reflect the genotype of XB201T strain in the iJO1366**, heterologous metabolic reactions **biosynthesizing D-phenyllactate** (*fldH* gene) were additionally introduced to the model and, **and flux values of native *E. coli* reactions were constrained to zero for the**

genes knocked out in the XB201T strain. For the implementation of *in silico* flux response analysis, the flux values of each central and aromatic amino acid biosynthesis reaction were constrained from their minimum values to the maximum values. D-phenyllactic acid production rate (D-lactate dehydrogenase) was subsequently maximized as an objective function at each constraint point where flux values of each central and aromatic amino acid biosynthesis reaction were fixed. The relationships between D-phenyllactic acid production rate and fluxes of each central and aromatic amino acid biosynthesis reaction were categorized into six types based on the flux patterns obtained from the *in silico* flux response analysis. Reactions that were predicted to be negatively correlated with D-phenyllactic acid production rate were identified as potential knockout candidates.”

11. The figures have no numbers.

[RESPONSE] This was due to the pdf generation process during the submission, which will be resolved during the final publication process.

12. Fig. 3. The structures most likely should represent the methylated form of poly(3HB-co-D-phenyllactate), but on C1 the methyl group should be replaced by methoxy group.

[RESPONSE] Actually, the structures presented in our original manuscript are correct. Since the ester bond is formed between the carboxylic group and alcohol group, the current presentations are correct.

13. Fig. S1. The negative results should be combined with those of Fig. S7.

[RESPONSE] We do not understand this comment. Figure S1 shows the negative results obtained by employing Pct, while Figure S7 shows the results of employing newly found HadA. Combining these two figures is not a problem for us to do, but we think that separation of two figures is much better for the readers to follow the paper.

14. Figs. S2 and S3: Explain that the small peak at $m/z + 22$ represents the monosodium form.

[RESPONSE] Thank you. The small peak at $m/z +22$ is the monosodium form of the corresponding CoA moiety. It is synthesized during the *in vitro* reaction. Since sodium acetyl-CoA was used as a substrate in the *in vitro* reaction, monosodium forms of the corresponding CoA moieties were generated. As the reviewer kindly suggested, we added the following sentence in the legends of Supplementary Figure 2 and 3.

Supplementary Information, Page 4, line 7; “The monosodium cinnamoyl-CoA was also detected.”

Supplementary Information, Page 5, line 5; “The monosodium phenyllactyl-CoA and monosodium 4-hydroxyphenyllactyl-CoA were also detected.”

15. Fig. S5a. The structure represents 1-phenyl-2-methoxy-3-oxobutane but not methyl phenyllactate.

[RESPONSE] Thank you. We made a mistake, which is now corrected. Thank you again for pointing this out.

16. Fig. S5b. In the 4-hydroxyphenyllactate moiety on C1 the methyl group should be replaced by methoxy group.

[RESPONSE] This is the same comment as the comment #12. The structure shown in the original manuscript is correct. Also, see our response to #12.

Reviewer #3 (Remarks to the Author):

Authors reported a one-step fermentative production of aromatic polyesters from glucose by metabolically engineered *Escherichia coli* strains expressing a newly identified coenzyme A (CoA) transferase and an evolved polyhydroxyalkanoate (PHA) synthase. Isocaprenoyl-CoA:2-hydroxyisocaproate CoA-transferase (HadA) capable of generating various aromatic hydroxyacyl-CoAs using acetyl-CoA as a CoA-donor. This is something really new and innovative. The paper can be accepted after following minor issues are properly addressed:

[RESPONSE] Thank you very much for appreciating the importance of our work.

1. Page 6, lines 2-3 "4CL and FldA can potentially be used for phenyllactyl-CoA generation and consequently aromatic polyester production". The 4CL and FldA are two key enzymes for phenyllactyl-CoA generation, authors are suggested to provide a little bit more info on these two enzymes even though they have supplementary info (most readers do not usually go to see the supplementary info).

[RESPONSE] Thank you very much. As the reviewer suggested, we added more detailed information on 4CL and FldA in the Result section so that the reader can easily understand.

Page 6, lines 1-2;

“As expected, phenyllactyl-CoA was successfully synthesized by *in vitro* sequential reaction

of 4CL and FldA; 4CL converts cinnamate to cinnamoyl-CoA, and then FldA converts phenyllactate into phenyllactyl-CoA (Supplementary Fig. 3).”

2. The newly synthesized aromatic PHAs look interesting. Are they random or block copolymers? The way authors described their structures, they are block copolymers, yet authors said that they are random copolymer. The NMR description is not clear to me. Please clarify them based on the NMR, better with a DSC graph.

[RESPONSE] Actually, the structures we have shown in the original manuscript are random copolymers as we do not have [monomer 1]_x [monomer 2]_y type presentation. Also, 2D NMR spectrum suggests that the synthesized polymers are random copolymers rather than block copolymers. We added “random copolymer” in the corresponding sentences.

3. The aromatic PHA containing 3HB is new in the area. The authors should provide some characterizations on molecular weights, thermal property et al although mechanical property is difficult at this stage due to the small amount of materials available.

[RESPONSE] We examined these material properties/characteristics already and presented them in Supplementary Table 1 and 2 in the original manuscript.

4. Page 9, lines 23-25 "poly(55.2 mol% 3HB-co-43.0 mol% D-phenyllactate-co-1.8 mol% D-mandelate)". This is also a new PHA. However, I would assume this is a very sticky material without a clear application area. Please elaborate.

[RESPONSE] Thank you very much. We measured T_g of poly(55.2 mol% 3HB-co-43.0 mol% D-phenyllactate-co-1.8 mol% D-mandelate), which was 24.9 °C. Also, the purified polymer was not sticky and was much more rigid than poly(3HB). There has been a report on chemical synthesis of polymandelate, a homopolymer of mandelate, which was suggested to be a thermo-resistant degradable polymer similar to polystyrene having relatively high T_g of ~100 °C (Reference). Poly(55.2 mol% 3HB-co-43.0 mol% D-phenyllactate-co-1.8 mol% D-mandelate) produced in this study is expected to possess material properties in between polypropylene and polystyrene.

Reference; Liu, T., Simmons, T. L., Bohnsack, D. A., Mackay, M. E., Smith, M. R., & Baker, G. L. Synthesis of polymandelide: A degradable polylactide derivative with polystyrene-like properties. *Macromolecules*, **40**, 6040-6047 (2007).

5. The PHA content in cell dry weight (CDW) is not high in this study, maybe the mutated PhaC has a low activity. Could authors find a way to increase PHB content? for example, by using a stronger promoter for phaCAB? or by optimizing RBS?

[RESPONSE] Thank you for the important comment. We assume that the reviewer was asking “PHA content”, rather than PHB content. As you mentioned, reinforcing the expression levels of *phaAB* genes by using strong promoter is enough to increase the PHA content as done in this work. According to the reviewer’s comment, we performed additional production studies. We responded to the comment 2 of Reviewer #1 (see above). In summary, according to your kind suggestion, we performed additional strain engineering and fed-batch cultivations to produce 13.9 g l⁻¹ of poly(61.9 mol% 3HB-co-38.1 mol% D-phenyllactate) with a polymer content of 55.0 wt% from glucose by fed-batch fermentation (Supplementary Fig. 9). Thank you for your nice comment to include higher titer (13.9 g l⁻¹) of aromatic polyester through additional experiments, which made this paper further improved.

6. The novelty of this paper is "the development of a bacterial platform system that allows production of various aromatic polyesters by one-step fermentation from glucose". Could authors make aromatic monomers from glucose for polymerization? This could generate better polymer properties. Could authors discuss this possibility?

[RESPONSE] Thank you. Actually, this is one of the key accomplishments of this study. We designed and reconstructed new metabolic pathways for the production of several aromatic monomers including D-phenyllactate and D-mandelate directly from glucose. Then, these monomers were polymerized into aromatic polyesters in this study by integrating monomer generation pathway with polymerization machinery, which resulted in “one-step direct fermentative production of aromatic polyesters from glucose”. These results were already described in the original manuscript. This strategy can be further extended to generate more diverse monomers.

Reviewers' Comments:

Reviewer #1 (Remarks to the Author):

The authors have significantly improved the quality of the paper. The comments have been addressed and the importance is articulated.

Reviewer #2 (Remarks to the Author):

The authors made minimal efforts necessary to meet the comments of the reviewers.

Answers to the comments of reviewer 2:

1. The authors still express proteins!
3. The authors refused to use the correct name of propionate CoA-transferase. Propionyl-CoA transferase is the wrong name, even if others use it. See. Ref 9 of the supplement: "Yang, T. H. et al. Biosynthesis of polylactic acid and its copolymers using evolved propionate CoA-transferase and PHA synthase. *Biotechnol. Bioeng.* 105, 150-160 (2010)." The authors also write: 2-hydroxyisocaproate CoA-transferase.
5. "We want to stick to rather than glucose (20 g l⁻¹) as this is the style we used throughout the manuscript." 20 g l⁻¹ glucose reads as if glucose rather than water is the solvent. Why don't the authors use mM, which is independent whether the formula of glucose is C₆H₁₂O₆ or C₆H₁₂O₆ x H₂O? Writing "111 mM glucose" is correct.
11. The figures have still no numbers.
12. The structures look like methylated after the brackets; indicate the continuing polymer differently. The same is valid for Fig. S5b. But the structure of Fig. S5a is phenyllactic acid methylether not ester; also Fig. S10c mandelic acid methylether.
13. Sorry for the comment, answer is OK.

Further comment:

Fig. 2C. In the structure of 4-hydroxybenzoyl-CoA a CO is missing, but the mass is correct.

Reviewer #3 (Remarks to the Author):

Authors have addressed my concerns well, with additional experiments and discussion. The revised paper should be accepted for publication now.

Responses to the Comments

Editor's Comments

Dear Prof Lee,

Your manuscript entitled "Microbial production of aromatic polyesters from glucose" has now been seen again by our previous referees, whose comments appear below. In light of their advice I am delighted to say that we are happy, in principle, to publish a suitably revised version in Nature Communications under the open access CC BY license (Creative Commons Attribution v4.0 International License). We therefore invite you to revise your paper one last time to address the remaining concerns of our reviewers. At the same time we ask that you edit your manuscript to comply with our format requirements and to maximise the accessibility and therefore the impact of your work.

[Response]: Thank you very much. Reviewers # 1 and #3 suggested to accept our revised manuscript. Reviewer #2 had some more minor comments, which were helpful in thoroughly revising our manuscript to its final form.

Your paper will be accompanied by a two-sentence editor's summary, of between 250-300 characters, when it is published on our homepage. Could you please approve the draft summary below or provide us with a suitably edited version.

[Response]: Thank you very much. The summary you wrote is GREAT! Only the strain name was changed to italic as shown in red.

Fermentative production of aromatic polyesters from glucose has been unsuccessful. Here, the authors achieve the objective by one-step fermentation of metabolically engineered *E. coli* expressing a new isocaprenoyl-CoA:2-hydroxyisocaproate CoA-transferase and an evolved polyhydroxyalkanoate synthase.

Reviewers' comments:

Reviewer #1 (Remarks to the Author):

The authors have significantly improved the quality of the paper. The comments have been addressed and the importance is articulated.

[Response]: Thank you very much for your kind effort of reviewing and improving our original manuscript.

Reviewer #2 (Remarks to the Author):

The authors made minimal efforts necessary to meet the comments of the reviewers.

[Response]: We do not agree with this comment. We did our best to improve polymer production through many additional experiments as suggested by the reviewers, and revised the manuscript. The other two reviewers were satisfied with the revision.

1. The authors still express proteins!

[Response]: Thank you for the comment. We changed “expression of proteins” to “expression of genes”.

3. The authors refused to use the correct name of propionate CoA-transferase. Propionyl-CoA transferase is the wrong name, even if others use it. See. Ref 9 of the supplement: “Yang, T. H. et al. Biosynthesis of polylactic acid and its copolymers using evolved propionate CoA-transferase and PHA synthase. Biotechnol. Bioeng. 105, 150-160 (2010).” The authors also write: 2-hydroxyisocaproate CoA-transferase.

[Response]: Corrected as suggested.

5. “We want to stick to rather than glucose (20 g l⁻¹) as this is the style we used throughout the manuscript.” 20 g l⁻¹ glucose reads as if glucose rather than water is the solvent. Why don’t the authors use mM, which is independent whether the formula of glucose is C₆H₁₂O₆ or C₆H₁₂O₆ x H₂O? Writing “111 mM glucose” is correct.

[Response]: As you suggested, we changed the concentrations of glucose and others as “g l⁻¹ of” throughout the manuscript. For the first concentration appearing for glucose, we also showed mM concentration in parenthesis.

11. The figures have still no numbers.

[Response]: As we responded before, this is what happens when we separately upload figures and an automatic merged pdf is generated – no problem at all in final publication.

12. The structures look like methylated after the brackets; indicate the continuing polymer differently. The same is valid for Fig. S5b. But the structure of Fig. S5a is phenyllactic acid methylether not ester; also Fig. S10c mandelic acid methylether.

[Response]: Thank you for pointing out our mistake. Much appreciated. Asterisks were added to show that the structure represents a monomer in the polymer. Again, thank you for pointing out the wrong structure of phenyllactate methylester. We now corrected it.

13. Sorry for the comment, answer is OK.

[Response]: Thank you.

Further comment:

Fig. 2C. In the structure of 4-hydroxybenzoyl-CoA a CO is missing, but the mass is correct.

[Response]: Thank you very much for thoroughly checking these. Again, we made a mistake, which is now corrected.

Reviewer #3 (Remarks to the Author):

Authors have addressed my concerns well, with additional experiments and discussion. The revised paper should be accepted for publication now.

[Response]: Thank you very much for your kind effort of reviewing and improving our original manuscript.